# Impact of School and Peer Connectedness on Adolescent Mental Health and Well-Being Outcomes during the COVID-19 Pandemic: A Longitudinal Panel Survey

**DOI:** 10.3390/ijerph19116768

**Published:** 2022-06-01

**Authors:** Emily Widnall, Lizzy Winstone, Ruth Plackett, Emma A. Adams, Claire M. A. Haworth, Becky Mars, Judi Kidger

**Affiliations:** 1Population Health Sciences, Bristol Medical School, University of Bristol, Bristol BS8 2PS, UK; lizzy.winstone@bristol.ac.uk (L.W.); becky.mars@bristol.ac.uk (B.M.); judi.kidger@bristol.ac.uk (J.K.); 2Division of Primary Care and Population Health, University College London, London NW3 2PF, UK; ruth.plackett.15@ucl.ac.uk; 3Population Health Sciences Institute, Newcastle University, Newcastle upon Tyne NE3 4ES, UK; emma.adams@newcastle.ac.uk; 4School of Psychological Science, University of Bristol, Bristol BS8 1TU, UK; claire.haworth@bristol.ac.uk

**Keywords:** mental health, well-being, social connectedness, adolescents, school, COVID-19, lockdown

## Abstract

School closures and social distancing measures during the pandemic have disrupted young people’s daily routines and social relationships. We explored patterns of change in adolescent mental health and tested the relationship between pre-pandemic levels of school and peer connectedness and changes in mental health and well-being between the first lockdown and the return to school. This is a secondary analysis of a longitudinal 3-wave panel survey. The study sample included 603 students (aged 13–14) in 17 secondary schools across south-west England. Students completed a survey pre-pandemic (October 2019), during lockdown (May 2020) and shortly after returning to school (October 2020). Multilevel models, with random effects, were conducted for anxiety, depression and well-being outcomes with school and peer connectedness as predictor variables. Symptoms of anxiety decreased from pre-pandemic to during the first UK lockdown and increased on the return to school; anxious symptoms decreased the most for students reporting feeling least connected to school pre-pandemic. Students reporting low levels of school and peer connectedness pre-pandemic experienced poorer mental health and well-being at all time points. Low school connectedness pre-pandemic was associated with a greater increase in anxious and depressive symptoms between lockdown and the return to school when compared to students with medium levels of school connectedness. No associations were found with high school connectedness or with low/high peer connectedness. For adolescents with poor school connectedness, the enforced time away from school that the pandemic caused led to reduced anxiety. Going forwards, we need to consider ways in which to promote connection with school as a way of supporting mental health and well-being.

## 1. Introduction

The COVID-19 pandemic and restrictions implemented by the UK government have substantially disrupted the lives and daily routines of the population. The effect of the lockdowns, school closures and social distancing measures on adolescent mental health and well-being is a particular area of concern. We have seen continually rising levels of mental health difficulties in adolescents, and there are concerns that pandemic-related challenges will have worsened an already concerning picture [1].

Repeated school closures, self-isolation policies and ongoing restrictions in schools are likely to have had a considerable impact on young people’s mental health and well-being over the course of the pandemic, particularly given that school routines are important coping mechanisms for young people with mental health issues or those with special educational needs [2]. Schools can also act as a ‘de facto’ mental health system for many children and adolescents [3]. School closures are therefore likely to be particularly disruptive for those who were seeking help through school systems.

Early studies have indicated relatively high levels of mental health symptoms during the pandemic [4], and the recent NHS Digital survey of children and young people’s mental health in England [1] highlighted that in July 2020—at the end of the first national lockdown—the proportion of children and young people with a probable mental health disorder was one in six, compared to one in nine in 2017. However, it is difficult to say how much of this is due to the pandemic as levels of mental health were already rising. Only five US studies to date have examined longitudinal change with available mental health data pre-pandemic, two of which relied on parent report data. Three studies found increases in depression and anxiety outcomes [5,6,7]. One found no meaningful effects of the pandemic on mental health outcomes, and a final study focusing on early adolescents found reductions in mental health outcomes, particularly for youths who had elevated levels of mental health problems before the pandemic [8].

Existing research has also demonstrated that the impact of the pandemic has differed by age groups. Parents of children in England have reported increased emotional symptoms in younger children (4–10 years), whilst adolescents (11–17 years) have seen a decrease in emotional symptoms [9]. An Italian study found longitudinal increases in childhood depression symptoms in children aged 7–12 years between April–June 2020 [10]. A recent qualitative study of young people in the UK (16–19 years) also highlighted some positive affordances of lockdown, including relief from academic pressures, time for personal development and strengthened relationships [11]. A survey study of 306 Italian adolescents (15–21 years) also revealed adaptive coping strategies during April 2020, including developing new interests [12]. A longitudinal study in New York found decreasing trajectories of depression and anxiety from May–July 2020 in adolescents and young adults (12–22 years) [13]. It is likely that adolescents are likely better able to stay meaningfully connected with their peers online in comparison to younger children and may be more capable of self-directed learning whilst away from school.

The impact of the pandemic on young people’s mental health is still an emerging area of research, with little evidence available to examine longitudinal change from pre-pandemic levels. The majority of studies to date are either cross-sectional or involve online convenience samples that demonstrate change in mental health and well-being during the course of the pandemic [9,14]. Studies have also used historical data to predict mental health outcomes during the pandemic [15] as well as independent samples to provide pre-pandemic comparisons [16]. Although these studies have provided valuable early insights, the convenience and non-probability samples are likely to contain substantial bias [17].

One area that has yet to be investigated is the impact of social connectedness on young people’s mental health and well-being outcomes during the pandemic. Young people’s relationships to their schools and peers are of particular interest, given lengthy school closures and various social distancing measures leading to a lack of in-person interaction with peers [18]. Stemming from belongingness theory, the construct of social connectedness refers to feelings of affiliation associated within a social network [19] and reflects the perceived feelings of meaningful connection with others at an interpersonal level [20]. The construct of social connectedness has been found to be a key determinant of adolescent mental health and well-being [21,22]. Social connectedness plays a central role in adolescence, and studies have shown the impact of peer relationships and social isolation to be risk factors for depression [23]. Equally, peer relationships can act as a protective factor against school disengagement among at-risk adolescents [24] as well as protect against mental health problems and strengthen adolescent resilience more generally [25]. The literature indicates the increased importance of peer interactions during adolescence, including spending more time with peers [26], the increased importance of peer social approval [27] and adolescents being more sensitive to peer acceptance, rejection and approval [28,29,30]. A recent viewpoint highlights concerns that physical distancing may have a disproportionate effect on adolescents due to the vital nature of peer interactions on their development [31]. There is much less research, however, on the relation between school connectedness and mental health symptoms in adolescents, but research indicates associations between school connectedness and adolescent depressive symptoms and a predictive link from school connectedness to future mental health problems [32,33].

A study of 423 11–15 year olds during the pandemic in India found 56% of young people were concerned about not attending school and 80% were concerned about not being able to see friends [34]. A study of 451 adolescents and young adults also found that greater COVID-19 school concerns were uniquely associated with increased depression symptoms [6]. Previous studies have also revealed that deficits in school connectedness are associated with more severe emotional symptoms such as depression [35] and higher levels of school connectedness are associated with positive adaptation and being less negatively influenced by stressful experiences [36].

Given the current knowledge gap surrounding the impact of social connectedness on young people’s mental health and well-being during the pandemic, this study aims to specifically explore the role of pre-pandemic levels of school and peer connectedness on adolescent mental health and well-being outcomes during the pandemic. We have previously reported on descriptive changes in mental health and well-being in adolescents between pre-pandemic and during the first UK lockdown (May 2020) [37]. This paper extends these findings to show patterns across three timepoints to include when schools first fully reopened. Our analysis focuses more specifically on changes in mental health and well-being between the first UK lockdown and the return to school.

### 1.1. Aims


To explore patterns of change over time in symptoms of anxiety, depression and well-being scores from pre-pandemic, during the first lockdown and when schools first fully reopened.To describe whether patterns of change over time differ according to pre-pandemic levels of school and peer connectedness.To test the relationship between pre-pandemic levels of school and peer connectedness and changes in mental health and well-being scores between lockdown and on the return to school.


### 1.2. Hypotheses

Aim one and two were exploratory and aim three was hypothesis-driven. We hypothesised that:Students with low levels of school and peer connectedness pre-pandemic would experience a greater increase in anxious and depressive symptoms, and a greater reduction in well-being scores on the return to school compared to students with medium levels of pre-pandemic connectedness.Students with high levels of school and peer connectedness would experience a greater decrease in anxious and depressive symptoms, and a greater increase in well-being scores on the return to school compared to students with medium levels of pre-pandemic connectedness.

## 2. Materials and Methods

### 2.1. Study Design

This study is a secondary analysis of a longitudinal three-wave survey study. Data was collected at three time points: wave 1: October 2019 (pre-pandemic); wave 2: May 2020 (during lockdown) and wave 3: October 2020 (1 month after schools first reopened). All year 9 students (aged 13–14 years) were invited to complete an online survey [38] during lesson time. For wave 2 (during lockdown), students were invited by their schools to complete the online survey at home. Paper surveys were provided for schools on request.

### 2.2. Participants

For the original study, all non-fee-paying secondary schools across four local authorities in south-west England (*n* = 76) were invited to participate. The secondary analysis presented here includes data from the 17 schools that participated at all three time points. Participating schools were broadly comparable to non-participating schools in the target area (see Appendix A for further information). All year 9 students (aged 13–14 years) in participating schools were invited to complete the survey at baseline during lesson time. At time 2 (during lockdown), the same students were asked to complete the survey online at home. At time 3, students who were in year 10 (aged 14–15 years) were asked to complete the survey at school during lesson time, or at home if unable to attend school. Data at all three time points were available for 603 participants.

Mean age at baseline was 13.2 years, 59.6% of the sample were female and 81.9% of the sample were White/Caucasian. The majority of students in the sample did not receive free school meals (93.7%) and the mean FAS index score was 9.23, indicating relatively high socio-economic status across the sample, with only 9.5% of students falling into the ‘low affluence’ category.

The matched analytic sample is described in Table 1. Compared with participants for whom complete data were not available, Black, Asian and minority ethnic students, students receiving free school meals, students reporting low family affluence and students at risk of depression were under-represented in the analytic sample.

### 2.3. Predictor Variables

School connectedness was measured using six items adapted from the Psychological Sense of School Membership Scale (PSSM) [39] and the School Connectedness Scale [40], as described in Jose et al. [21]. The scale (time 1 α = 0.87) includes three items assessing student relationships with teachers (e.g., “I always get an opportunity to talk with my teacher(s)”) and three sense of-school community items (e.g., “I feel proud about my school”). The six items were scored on a 5-point scale ranging from 1 (“strongly disagree”) to 5 (“strongly agree”). School connectedness scores were computed by summing the six items, with higher scores indicating greater connectedness (time 1 α = 0.88).

Peer connectedness was measured using a 7-item sub-scale [21] (time 1 α = 0.80). Two school peer relationship items asked how well students got on with their classmates and other students in the school. Item response options ranged from 1 (“not at all well”) to 5 (“really well”). Two items measured happiness with number of close friends and used a 5-point scale ranging from 1 (“very unhappy”) to 5 (“very happy”). Three peer support items (e.g., “I can trust my friends with personal problems”) used a 5-point Likert scale. Peer connectedness scores were computed by summing the six items, with higher scores indicating greater connectedness.

### 2.4. Outcome Measures

The 14-item Hospital Anxiety and Depression Scale (HADS) was used to measure depression symptoms (7-items) and anxiety (7-items) at each time point. Each item is coded from 0 (not at all) to 3 (nearly all the time), and scores for anxious symptoms (time 3 α = 0.88) and depressive symptoms (time 3 α = 0.78) can therefore range from 0–21. Items include “I feel tense or wound up” and “I feel as if I am slowed down”. HADS has been validated with 12–17 year olds [41].

The 14-item Warwick and Edinburgh Mental Well-Being Scale (WEMWBS) was used to measure well-being at each time point (time 3 α = 0.93). Scores range from 14 to 70. Items include “I’ve been feeling optimistic about the future” and “I’ve been feeling cheerful”. Response options range from 0 (none of the time) to 5 (all of the time). WEMWBS has been validated for use with children from the age of 13 and over [42].

Confounders: Demographic variables included gender (male and female), ethnicity (White and Black, Asian and minority ethnic), sexual orientation (heterosexual and LGBTQI+) and disability (no disability and limiting long-term illness or disability). Socioeconomic status (SES) was measured by the six-item Family Affluence Scale III [43]. All demographic variables were collected via self-report within the survey.

### 2.5. Statistical Analysis

Matched data across all three time points were available for 603 students. Complete case analysis was conducted and all data was analysed in Stata 15.

Wilcoxon signed-rank tests were performed to test whether changes over time in anxious and depressive symptoms and well-being scores were statistically significant. We also mapped the changes in mental health and well-being scores over time by levels of school and peer connectedness at baseline (pre-pandemic).

Multilevel models, with random effects at the school and student levels to account for clustering and repeated measures, respectively, were used to investigate the association between pre-pandemic levels of school/peer connectedness and changes in mental health and well-being outcomes between wave 2 (during lockdown) and wave 3 (return to school). Separate multilevel models were used for each outcome (depressive symptoms, anxiety symptoms and well-being scores) and each predictor variable (school connectedness and peer connectedness) creating 6 models in total.

Models were adjusted for several potential confounding variables based on a priori grounds and previously published adolescent mental health literature, including ethnicity, socioeconomic status (family affluence), sexual attraction and limiting long-term illness of disability. Unadjusted models included only the predictor variable (school or peer connectedness) and outcome measure (depressive symptoms, anxious symptoms and well-being scores), partially adjusted models included the predictor variable, outcome measure and demographics (gender, ethnicity, sexual orientation, family affluence, disability) and the fully adjusted model included the predictor variable, outcome measure, demographics and the relevant baseline connectedness measure (for example, if school connectedness was the predictor variable, the fully adjusted model included wave 1 peer connectedness). Baseline peer and school connectedness were found to be moderately correlated (r = 0.36).

Due to theoretically driven hypotheses concerning the relationship between low and high school connectedness and mental health and well-being outcomes, both connectedness variables were categorised into three groups (low, middle and high), using the 25th, 50th and 75th percentiles of each variable. We predicted distinct differences in mental health and well-being outcomes on the return to school for students reporting low and high school/peer connectedness pre-pandemic and therefore used the ‘middle’ connectedness group as the reference category.

We used maximum likelihood for more accurate estimation of standard errors. Change scores from time 2 (during lockdown) and time 3 (return to school) were calculated for symptoms of anxiety and depression (HADS) and well-being scores (WEMWBS) and used within the multilevel models.

#### Missing Data

Of the 603 students who completed the survey at all three timepoints, levels of data completeness were very high across both predictor and outcome variables. Table 2 shows the percentage of complete data for each predictor and outcome variable.

## 3. Results

Wilcoxon signed-rank tests showed a statistically significant decrease in anxious symptoms between pre-pandemic and the first UK lockdown (z = −7.061, *p* =< 0.001) and a statistically significant increase in anxious symptoms between lockdown and the return to school (z = 6.983, *p* =< 0.001). There were no significant changes over time for depressive symptoms either between pre-pandemic to lockdown (z = −0.319, *p* = 0.750) or from lockdown to the return to school (z = 1.614, *p* = 0.107). There was a significant increase in well-being scores from pre-pandemic to during lockdown (z = 3.847, *p* =< 0.001) and a significant decrease in well-being scores from during lockdown to return to school (z = −3.654, *p* = 0.003).

Figure 1 charts the change in students’ anxious and depressive symptoms and well-being scores over the three waves of the study according to pre-pandemic levels of school and peer connectedness. Across all three outcomes, estimates were consistently poorest among students who reported feeling least connected to their school and peers before the pandemic.

Table 3 reports the association between pre-pandemic levels of school connectedness and change in anxious and depressive symptoms and well-being scores between lockdown (May 2020) and the return to school (October 2020). In the fully adjusted models, low school connectedness at baseline was associated with a greater increase in anxious symptoms (β = 1.332, *p* = 0.001, 95% CI [0.55, 2.11]) and depressive symptoms (β = 0.869, *p* = 0.010, 95% CI [0.21, 1.53]) between lockdown and the return to school, when compared to students with medium levels of school connectedness. Although well-being scores decreased from lockdown to the return to school for students reporting low school connectedness as baseline, this was a smaller effect, with 95% confidence intervals just crossing 0 (β = −1.91, *p* = 0.056, 95% CI [−3.87, 0.05]). There was no evidence to suggest a difference in change scores for those with high compared to medium levels of connectedness for anxious symptoms (β = 0.203, *p* = 0.602, 95% CI [−0.56, 0.96]), depressive symptoms (β = 0.74, *p* = 0.590, 95% CI [−0.79, 0.45]) or well-being scores (β = −0.74, *p* = 0.436, 95% CI [−2.58, 1.11]).

Table 4 reports the association between pre-pandemic levels of peer connectedness and the changes in anxious and depressive symptoms and well-being scores between lockdown (time 2) and the return to school (time 3). Low peer connectedness was associated with an increase in anxious symptoms within the partially adjusted model (β = 0.756, *p* = 0.049, 95% CI [−0.01, 1.51]); however, findings attenuated to the null following adjustment for school connectedness (β = 0.531, *p* = 0.175, 95% CI [−0.24, 1.30]). Consistent with the findings for school connectedness, we did not find differences in outcome change scores for students with high compared to medium levels of peer connectedness for anxious symptoms (β = 0.100, *p* = 0.798, 95% CI [−0.67,0.87]), depressive symptoms (β = −0.232, *p* = 0.473, 95% CI [−0.87, 0.40]) or well-being scores (β = 0.457, *p* = 0.636, 95% CI [−1.43, 2.34]).

## 4. Discussion

To date, this is the first study in the UK to explore longitudinal change in early adolescent mental health and well-being with available baseline mental health data pre-pandemic.

Findings from descriptive data suggest that levels of mental health and well-being were consistently poorest among students who reported feeling least connected to their school and peers before the pandemic. Changes over time were most prominent for symptoms of anxiety with a decrease during lockdown and a subsequent increase on the return to school. The most striking change was the decrease in anxiety for students reporting feelings least connected to school pre-pandemic.

In the regression analysis, after adjusting for potential confounders, low school connectedness was associated with an increase in anxious and depressive symptoms between lockdown and the return to school. Findings for well-being were very similar with well-being scores decreasing for students reporting low school connectedness, but there was a less strong effect. No association was found for high school connectedness, or for low/high peer connectedness. Our hypotheses were therefore only partially supported.

The observed overall decrease in symptoms of anxiety during lockdown is consistent with previous work demonstrating a reduction in adolescent emotional symptoms during early lockdown [14] as well as some young people reporting benefits from lockdown [44]. This reduction in anxiety also aligns with a recently published study that found one-third of young people (8–18 years) reported improved mental well-being during lockdown [45] as well as the NHS Digital Survey, which reported 27% of 11–16 year olds felt that lockdown had made their lives better [1].

Reductions in anxious symptoms during lockdown suggests that some students adapted well to the first school closures, which perhaps provided a chance for them to be away from the everyday pressures of the school environment. Existing research demonstrates a number of aspects of the school environment that are linked to poorer mental health, including bullying [46] and academic stress [47]. Additionally, a recent US study of adolescent girls found that a positive impact on those found to have improved emotional health during lockdown was, reduced pressure from school [48]. Although we do not have data available to know whether levels of anxious symptoms fluctuated throughout the lockdown period, the increase in symptoms of anxiety on the return to school suggests an effect of school rather than time. However, this increase could also be an effect of starting a new school year.

Although we found overall decreases in anxiety across the sample, our findings around school connectedness suggest that these reductions in anxiety are likely to have been driven by those students feeling least connected to school. This study therefore suggests that being out for school for those not connected to school has a positive effect on depressive and anxious symptomatology.

The rise in anxious symptoms on the return to school may be due to school environment stressors being reintroduced, but could also indicate that communication about the return to school and available mental health support for students were insufficient. Data from the OxWell study highlighted that young people looked most forward to seeing friends again, but aspects causing young people concern related to school systems including lessons and homework [45]. It is also possible that the increase in anxious symptoms on the return to school is an appropriate reaction to changing circumstances during the pandemic and anxieties around new COVID-related rules, and perhaps these increases should only cause concern if they are sustained beyond the initial return phase. However, we did not observe symptoms of anxiety increasing above pre-pandemic levels, which we may expect if anxious symptoms were pandemic-specific.

Findings from this study build on the existing literature around the impact of school culture and environment on student mental health and well-being [49,50]. Students who reported feeling least connected to their school and peers pre-pandemic consistently reported poorer mental health and well-being at each time point. If some of the drivers of poor mental health and well-being lie within school and peer relationships at school, it makes sense for those who feel least connected to school to see improvements in mental health and well-being when removed from the school environment. These findings raise important questions about the role of the school environment on adolescent mental health and what it is that may be driving anxiety within the school environment, such as relationships with teachers, exam pressures or bullying. There is a need for qualitative research to address these questions.

Previous research examining the association between school connectedness and the school environment found that positive classroom management climates, participation in extracurricular activities, tolerant disciplinary policies and small school size were associated positively with higher school connectedness [51]. These factors are each amenable to change and could be areas of focus for school health promotion interventions. Our findings highlight that the concept of school health promotion should be expanded to be inclusive of creating school environments that make adolescents feel like they belong and are cared for at school.

It is likely that students who felt well connected to their school and peers pre-pandemic adapted well to lockdown and this may explain the lack of substantial changes in their mental health and well-being overtime. A recent commentary discusses the possibility that adolescents with robust, high-quality peer relationships may adapt better to lockdowns and might be better able to cope with the social restrictions [18]. This also supports previous findings indicating that young people with higher levels of school connectedness may adapt better and be less negatively influenced by stressful experiences [28].

The study offers insight into the importance of students feeling well connected to their school, including feeling respected and understood by teachers, having the opportunity to talk with teachers and enjoying attendance. However, we did not find that students with high school connectedness fared better on the return to school in terms of anxious and depressive symptoms or overall well-being. This finding suggests that the pandemic and new COVID-19 rules were anxiety-provoking for everyone, and the return to school after a long break was difficult even for those who were usually well connected to school. However, this study particularly draws attention to low school connectedness being problematic for mental health as it was this group with poorer mental health on the return to school.

This study raises an important question as to whether school connectedness directly helps mental health, or whether being at school whilst not feeling connected to your school provokes anxious and depressive feelings. The pandemic has offered a window of opportunity to explore this question, but more research is needed to examine this relationship further. A recent study highlights the importance of building school connectedness in view of COVID-19 and recommends the role of school nurses to help promote school connectedness and ensure students feel safe and supported [52].

Another interesting finding was the lack of association between peer connectedness and depressive symptoms or well-being scores, and the association between low peer connectedness and change in anxious symptoms attenuated after adjusting for school connectedness. This suggests that school connectedness was the main driver of this effect. This lack of association might be explained by the stability to peer relationships provided by online socialising in the absence of in-person interactions [31]. These findings support a recent US study that found an association between school connectedness and mental health even when accounting for social connectedness, suggesting that a student’s sense of connectedness to school contributes to mental health over and above their connection to peers [53].

We know from the existing literature that age is an extremely important factor in terms of how young people have experienced the pandemic. For some age groups, the first lockdown did not lead to a steep rise in mental health problems, but it is important to understand why this is. Existing studies have shown both older adolescents [44] and younger children [14] overall experienced poorer mental health, which differs from our findings on early adolescents. It is therefore important to remember that the pre-pandemic data was collected from students in Year 9, who were not undergoing substantial exam or transition pressures. It is also important to note that the data collected during lockdown was relatively early on in the pandemic (May 2020), when young people were unaware of the long-term nature of the pandemic. The rise in anxious symptoms on the return to school may reflect the experiences of the ongoing pandemic and multiple restrictions still in place several months later; however, symptoms of anxiety did not rise higher than baseline, which might be expected if there were additional pandemic-specific anxieties.

Further qualitative research is needed to understand the mechanisms leading to the changes in anxious symptoms observed in this study. Longitudinal data will be vitally important as the pandemic continues, particularly to see whether changes in anxious and depressive symptoms and well-being are observed in the longer term.

### Strengths and Limitations

The unique availability of matched individual level pre-pandemic mental health data is a major strength of this study, enabling changes in mental health and well-being to be observed over time. We also provide a novel focus on the impact of school and peer connectedness on mental health outcomes in the context of the pandemic and adjust for a range of potential confounding factors. However, there are some limitations to be taken into consideration. Firstly, the generalisability of our findings should be viewed with caution as Black, Asian and minority ethnic (BAME) students, and students from lower socioeconomic backgrounds were underrepresented in our analytic sample. The analytic sample also had fewer students at risk of depression (HADS score ≥7) at baseline compared to the sample for whom complete data were not available. Fee paying schools were also excluded from the original sample, restricting the extension of findings to students in private education. Another limitation that we are unable to discount is that improvements in mental health and well-being overtime also could be due to seasonal effects as opposed to due to lockdown.

Although the descriptive data suggest mental health and well-being is poorer overall for those with low school/peer connectedness, the direction of effects cannot be inferred. It is possible that mental health difficulties could result in someone feeling less connected to peers as well as vice versa. The data did not allow us to examine changes in connectedness over time.

Going forwards, prioritising routine longitudinal data collection will be a priority [54] to allow continued understanding of individual differences in varying age groups. It will be important to build up a rich data set as studies emerge in order to accurately capture the dynamic and multi-level factors affecting young people as the pandemic continues. The survey during lockdown was conducted within schools in south-west England towards the beginning of the pandemic and when the first national lockdown restrictions were beginning to ease (for example, students were allowed to meet friends outside). It will be important to understand how young people’s mental health has been impacted at different stages of the pandemic given changing restrictions. South-west England also had relatively low rates of COVID-19 at the time of the survey’s completion; it is therefore worth considering whether schools in this area experienced the pandemic differently compared to other parts of the UK. The mental health of young people in areas experiencing higher rates of morbidity and mortality may have been affected differently.

## 5. Conclusions

These findings point towards an important emerging area of research into school connectedness and its relationship to student mental health and well-being. Moving forwards, it will be important for schools and public health professionals to think about ways to measure how connected young people feel to their school in order to target interventions for those who have low levels of school connectedness. More research into how to improve students’ sense of connectedness is also required. These findings also provide evidence to support interventions to improve connectedness in young people, particularly improving how connected young people feel to their school. Regardless of levels of connectedness, symptoms of anxiety decreased during the first UK lockdown and then increased on the return to school, which suggests that something about school culture needs to change.

Few longitudinal studies investigating the impact of the COVID-19 pandemic have matched pre-pandemic data on adolescent mental health. This is the first UK study to date to examine changes in levels of adolescent mental health and well-being from pre-pandemic to school reopening. Additionally, no previous studies have longitudinally explored the impact of school and peer connectedness on mental health/well-being outcomes during lockdown and the return to school.

Although there may not be another national closure of schools, it is possible that some students may experience periods of home learning due to local school closures (to manage local outbreaks) or individual isolation. It is therefore important to enhance students’ connectedness to buffer these transitions and their impact on anxiety, mood and well-being.

## Figures and Tables

**Figure 1 ijerph-19-06768-f001:**
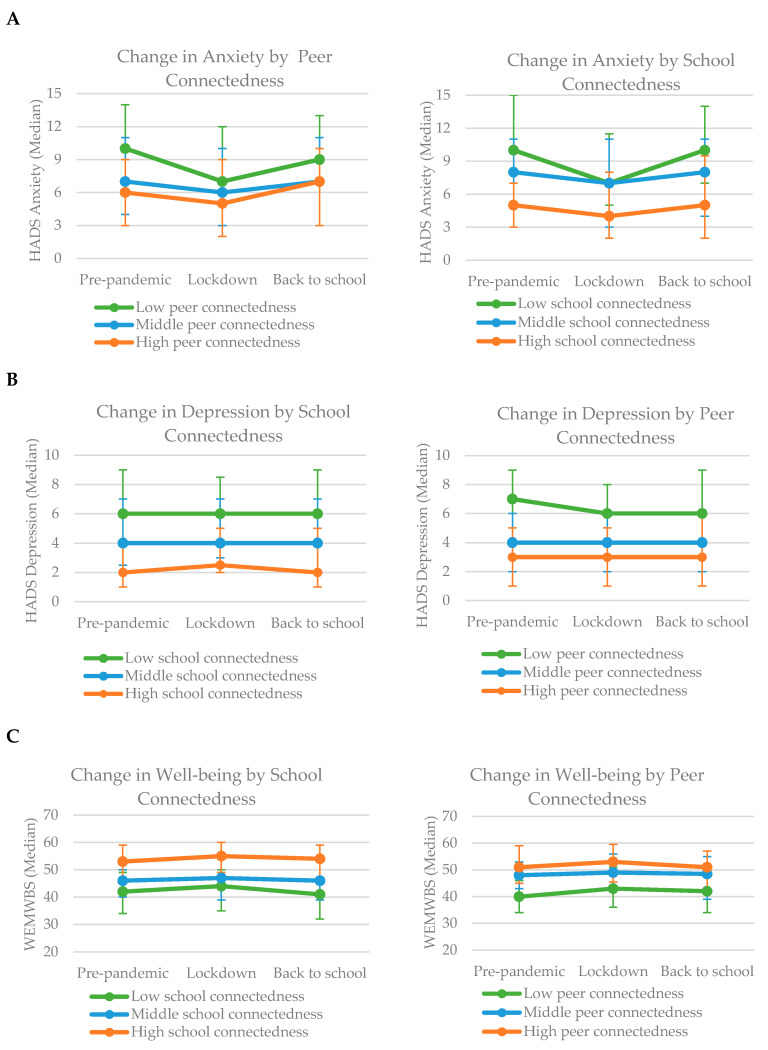
Change in mental health and well-being outcomes over time by levels of school and peer connectedness. (**A**) Change in Anxiety (HADS) over time by levels of school and peer connectedness. (**B**) Change in Depression (HADS) over time by levels of school and peer connectedness. (**C**) Change in Well-being (WEMWBS) over time by levels of school and peer connectedness.

**Table 1 ijerph-19-06768-t001:** Comparison of samples completing the survey at all three time points (analytic sample) and those with data available at fewer than three time points.

Comparison Variable	<3 Time Points *		All 3 Time Points **		Sample Comparison	
	*n*	% or Mean (SD)	*N*	% or Mean (SD)	Pearson Chi2	*p*-Value
Gender (% Female)	1613	56.1%	592	59.6%	2.19	0.139
Ethnicity (% White)	1648	76.9%	598	81.9%	6.59	0.010
Sexual attraction (% heterosexual)	1666	81.1%	599	81.4%	0.04	0.840
Free school meals (% receiving)	1609	11.7%	595	5.7%	17.38	<0.001
Family Affluence (% low affluence)	1630	15.9%	598	9.5%	15.06	<0.001
Reported a disability, health condition or SEN	1643	15.3%	599	13.8%	0.70	0.403
At risk of depression (HADS ≥ 7)	1684	32.0%	573	27.2%	4.59	0.032
At risk of anxiety (HADS ≥ 9)	1686	43.2%	577	41.2%	0.70	0.404

* Participants who completed fewer than three time points; ** Participants who completed all three time points of the survey (analytic sample).

**Table 2 ijerph-19-06768-t002:** Percentage of participants with complete data on exposure predictor and outcome measures by survey time point.

	Pre-Pandemic (T1, October 2019)	During Lockdown (T2, May 2020)	Return to School (T3, October 2020)
School connectedness	95.1%	96.5%	92.5%
Peer connectedness	95.3%	96.5%	93.0%
Depressive symptoms (HADS)	97.6%	97.3%	97.6%
Anxious symptoms (HADS)	97.5%	97.3%	97.6%
Well-being (WEMWBS)	98.3%	95.6%	93.3%

**Table 3 ijerph-19-06768-t003:** Association of school connectedness with change in adolescent anxious and depressive symptoms and well-being scores between lockdown and return to school.

**Anxious Symptoms (HADS) ^a^**	**Unadjusted**			**Partially Adjusted ***			**Fully Adjusted ****		
	**Est**	**95% CI**	***p*-Value**	**Est**	**95% CI**	***p*-Value**	**Est**	**95% CI**	***p*-Value**
Low school connectedness	1.409	(0.64, 2.17)	<0.001	1.414	(0.64, 2.19)	<0.001	1.332	(0.55, 2.11)	0.001
Middle school connectedness	0	Ref	-	0	Ref	-	0	Ref	-
High school connectedness	0.174	(−0.56, 0.91)	0.641	0.179	(−0.55, 0.91)	0.630	0.203	(−0.56, 0.96)	0.602
**Depressive Symptoms ^b^ (HADS)**	**Unadjusted**			**Partially Adjusted ***			**Fully Adjusted ****		
	**Est**	**95% CI**	***p*-Value**	**Est**	**95% CI**	***p*-Value**	**Est**	**95% CI**	***p*-Value**
Low school connectedness	0.953	(0.32, 1.59)	0.003	0.892	(0.25, 1.54)	0.007	0.869	(0.21, 1.53)	0.010
Middle school connectedness	0	Ref	-	0	Ref	-	0	Ref	-
High school connectedness	−0.184	(−0.79, 0.42)	0.548	−0.181	(−0.79, 0.42)	0.556	−0.17	(−0.79, 0.45)	0.590
**Well-Being Score (WEMWBS) ^c^**	**Unadjusted**			**Partially Adjusted ***			**Fully Adjusted ****		
	**Est**	**95% CI**	***p*-Value**	**Est**	**95% CI**	***p*-Value**	**Est**	**95% CI**	***p*-Value**
Low school connectedness	−1.898	(−3.81, 0.02)	0.053	−1.849	(−3.78, 0.08)	0.061	−1.91	(−3.87, 0.05)	0.056
Middle school connectedness	0	Ref	-	0	Ref	-	0	Ref	-
High school connectedness	−0.845	(−2.65, 0.97)	0.361	−0.813	(−2.61, 0.99)	0.376	−0.74	(−2.58, 1.11)	0.436

^a^ ICC = 4.51 × 10^−14^ (95% CI 4.51 × 10^−14^, 4.51 × 10^−14^); ^b^ ICC = 0.19 (95% CI 0.002, 0.148); ^c^ ICC = 0.12 (95% CI 0.001, 0.135); * partially adjusted: age, gender, ethnicity, sexual orientation, family affluence, disability; ** fully adjusted: age, gender, ethnicity, sexual orientation, family affluence, disability, baseline peer connectedness.

**Table 4 ijerph-19-06768-t004:** Association of peer connectedness with change in adolescent anxious and depressive symptoms and well-being scores between lockdown and return to school.

**Anxious Symptoms (HADS) ^a^**	**Unadjusted**			**Partially Adjusted ***			**Fully Adjusted ****		
	**Est**	**95% CI**	***p*-Value**	**Est**	**95% CI**	***p*-Value**	**Est**	**95% CI**	***p*-Value**
Low peer connectedness	0.902	(0.43, 1.37)	0.068	0.756	(0.01, 1.51)	0.049	0.531	(−0.24, 1.30)	0.175
Middle peer connectedness	0	Ref	-	0	Ref	-	0	Ref	-
High peer connectedness	0.037	(−0.72,0.80)	0.923	−0.053	(−0.08, 0.70)	0.891	0.100	(−0.67, 0.87)	0.798
**Depressive Symptoms (HADS) ^b^**	**Unadjusted**			**Partially Adjusted ***			**Fully Adjusted ****		
	**Est**	**95% CI**	***p*-Value**	**Est**	**95% CI**	***p*-Value**	**Est**	**95% CI**	***p*-Value**
Low peer connectedness	0.359	(−0.26, 0.98)	0.258	0.297	(−0.33, 0.92)	0.350	0.09	(−0.55, 0.73)	0.781
Middle peer connectedness	0	Ref	-	0	Ref	-	0	Ref	-
High peer connectedness	−0.354	(−0.99, 0.27)	0.266	−0.375	(−1.00, 0.25)	0.241	−0.232	(−0.87, 0.40)	0.473
**Well-being Score (WEMWBS) ^c^**	**Unadjusted**			**Partially Adjusted ***			**Fully Adjusted ****		
	**Est**	**95% CI**	***p*-Value**	**Est**	**95% CI**	***p*-Value**	**Est**	**95% CI**	***p*-Value**
Low peer connectedness	0.148	(−1.72, 2.02)	0.877	0.175	(−1.69, 2.03)	0.853	0.398	(−1.51, 2.31)	0.683
Middle peer connectedness	0	Ref	-	0	Ref	-	0	Ref	-
High peer connectedness	0.457	(−1.38, 2.34)	0.617	0.519	(−1.34, 2.38)	0.584	0.457	(−1.43, 2.34)	0.636

^a^ ICC = 3.22 × 10^−16^ (95% CI 3.22 × 10^−16^, 3.22 × 10^−16^); ^b^ ICC = 0.19 (CI 95% 0.002, 0.151); ^c^ ICC = 0.15 (CI 95% 0.002, 0.125); * partially adjusted: age, gender, ethnicity, sexual orientation, family affluence, disability; ** fully adjusted: age, gender, ethnicity, sexual orientation, family affluence, disability, baseline school connectedness.

## Data Availability

The data set is available on request.

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
