# Peer review of "Impact of School and Peer Connectedness on Adolescent Mental Health and Well-Being Outcomes during the COVID-19 Pandemic: A Longitudinal Panel Survey"

_ijerph, 2022, doi:10.3390/ijerph19116768_

Round 1

Reviewer 1 Report

Thanks for giving me the opportunity to review the paper “Impact of school and peer connectedness on adolescent mental health and well-being outcomes during the COVID-19 pandemic: a longitudinal panel survey” by Widnall et al.

The authors report the longitudinal changes in mental health and well-being in a sample of ~600 UK adolescents during the COVID-19 pandemic.  They observed poorer mental health and wellbeing among students who reported feeling least connected to their school and peers before the pandemic. 

Anxiety symptoms showed the most significant changes across time, with a decrease during the lockdown and a subsequent increase upon returning to school. This shift was particularly prominent in students who were less connected to their school.

This is a very interesting study which offers useful insights on the impact of school (and peer) connectedness on mental health in the context of the pandemic. Although the study is limited to a specific geographic area and adolescent group, as the authors clearly indicate in the discussion, I believe it is still important in the current literature. The study is well executed and methodologically sound, and it may be of particular interest to the IJERPH readers.

Please clarify the following:

The authors indicate a HADS score of > 6 as the cutoff value for depression risk in table 1 and section 4.1.  For both anxiety and depression, the optimal cutoff value for the HADS scale is said to be >=8.  Could the authors explain why a more relaxed value was chosen? Section 2.4 points to reference 39 for the validation of the scale, but I am unable to access the paywalled article and confirm whether that cutoff threshold is reported there. 

Minor typographical errors should be corrected in the text, such as missing spaces between words and citations, and inconsistent usage of brackets for citations (see for example page 10).

Author Response

The authors thank the reviewer for their positive comments on this manuscript.

The HADS cut-offs used are those suggested by White et al. 1999 due to their validation work with adolescents which matches the age group of this sample. "The lower cut-off for a possible case, 1,800 youths (30.9%) scored 9 or above for anxiety and 1,983 (34.0%) scored 7 or above for depression. "

These published cut-offs were used to identify adolescents at risk of either anxiety or depression in our sample. For clarity, I have revised Table 1 to state cut-off ≥9 for anxiety and cut-off ≥7 for depression (rather than >8 and >6).

The authors have now carried out a thorough proofread of the manuscript and corrected any typographical errors. Thank you for highlighting the inconsistencies in reference brackets and spaces.

Reviewer 2 Report

Congratulation to the authors for the nice work conducted. I really enjoy reading this document. The document is well structured and consistent with the object of research.
Introduction: the author/s provide data on the importance of expansion in this field of research.
Methodology: this section explains how the study was carried out and details the research design and measures used.
Results: this section explains the results obtained in an orderly and concise form, being easy to understand and consistent with what was stated in the theoretical framework.
Discussion: An analysis of the results is made and it is related to other studies so that the importance of the data obtained can be seen.
The author/s have made a good job, although I contribute here some suggestions for the improvement of the quality of the document: 
•    In the Discussion part: add more previous studies to confirm or not the results obtained. 
•    Explain more the possible practical applications of the study carried out.

Author Response

The authors thank the reviewer for their positive comments on this manuscript.

With reference to their suggested improvements, the following additional studies have now been added to the discussion:

Soneson E, Puntis S, Chapman N, Mansfield K, Jones P, Fazel M. Happier during lockdown: a descriptive analysis of self-reported wellbeing in 17,000 UK school students during Covid-19 lockdown. Eur Child Adolesc Psychiatry2022. https://doi.org/10.1007/s00787-021-01934-z

"This reduction in anxiety also aligns with a recently published study that found one-third of young people (8-18 years) reported improved mental well-being during lockdown [44]."

"Data from the OxWell study highlighted young people looked most forward to seeing friends again, but aspects causing young people concern related to school systems including lessons and homework [44]."

Arseneault L. Annual Research Review: The persistent and pervasive impact of being bullied in childhood and adolescence: implications for policy and practice. J Child Psychol Psychiatry. 2018, 59, 405-21.

Högberg B, Strandh M, Hagquist C. Gender and secular trends in adolescent mental health over 24 years–the role of school-related stress. Soc Scie Med. 2020, 250, 112890.

Silk JS, Scott LN, Hutchinson EA, Lu C, Sequeira SL, McKone KM, et al. Storm clouds and silver linings: Day-to-day life in COVID-19 lockdown and emotional health in adolescent girls. J Pediatr Psychol. 2022, 47, 37-48. 10.1093/jpepsy/jsab107

"We know of a number of aspects of the school environment are linked to poorer mental health including bullying [46] and academic stress [47]. A US study of adolescent girls found that a positive impact on those found to have improved emotional health during lockdown was reduced pressure from school [48]."

A further practical application has been added to the manuscript to include a recommendation of school nurses to enhance school connectedness:

McCabe EM, Davis C, Mandy L, Wong C. The role of school connectedness in supporting the health and well-being of youth: recommendations for school nurses. NASN School Nurse. 2022, 37, 42-7.

"A recent study highlights the importance of building school connectedness in view of COVID-19 and recommends the role of school nurses to help promote school connectedness and ensure students feel safe and supported [52]."

Reviewer 3 Report

Review: Impact of school and peer connectedness on adolescent mental 2 health and well-being outcomes during the COVID-19 pan-3 demic: a longitudinal panel survey.

Dear authors

The article  contains: very clear hypotheses, very good: conclusions and discussion.  I appreciate the structure of the study as well as your research,

I recommend this paper for publishing.

I suggest you to add studies of several authors, who´s work will expand (by reference) your study with other assumptions and observations that deal with the issue.

Tkáčová, H.; Pavlíková, M.; Tvrdoň, M.; Jenisová, Z. The Use of Media in the Field of Individual Responsibility for Sustainable Development in Schools: A Proposal for an Approach to Learning about Sustainable Development. Sustainability 2021, 13, 4138. https://doi.org/10.3390/su13084138 

Tkáčová, H., Pavlíková, M., Tvrdoň, M., Prokopyev, A.I.  Existence and prevention of social exclusion of religious university students due to stereotyping.  Bogoslovni Vestnik, 2021, 81(1), pp. 199–223.  https://www.teof.uni-lj.si/uploads/File/BV/BV2021/01/Tkacova.pdf

Author Response

The authors would like to thank the reviewers for their positive comments and we are pleased to hear that the manuscript is well structured and clear.

A number of additional studies have now been added to the manuscript to support findings and strengthen the manuscript:

Soneson E, Puntis S, Chapman N, Mansfield K, Jones P, Fazel M. Happier during lockdown: a descriptive analysis of self-reported wellbeing in 17,000 UK school students during Covid-19 lockdown. Eur Child Adolesc Psychiatry2022. https://doi.org/10.1007/s00787-021-01934-z

"This reduction in anxiety also aligns with a recently published study that found one-third of young people (8-18 years) reported improved mental well-being during lockdown [44]."

"Data from the OxWell study highlighted young people looked most forward to seeing friends again, but aspects causing young people concern related to school systems including lessons and homework [44]."

Arseneault L. Annual Research Review: The persistent and pervasive impact of being bullied in childhood and adolescence: implications for policy and practice. J Child Psychol Psychiatry. 2018, 59, 405-21.

Högberg B, Strandh M, Hagquist C. Gender and secular trends in adolescent mental health over 24 years–the role of school-related stress. Soc Scie Med. 2020, 250, 112890.

Silk JS, Scott LN, Hutchinson EA, Lu C, Sequeira SL, McKone KM, et al. Storm clouds and silver linings: Day-to-day life in COVID-19 lockdown and emotional health in adolescent girls. J Pediatr Psychol. 2022, 47, 37-48. 10.1093/jpepsy/jsab107

"We know of a number of aspects of the school environment are linked to poorer mental health including bullying [46] and academic stress [47]. A US study of adolescent girls found that a positive impact on those found to have improved emotional health during lockdown was reduced pressure from school [48]."

A further practical application has been added to the manuscript to include a recommendation of school nurses to enhance school connectedness:

McCabe EM, Davis C, Mandy L, Wong C. The role of school connectedness in supporting the health and well-being of youth: recommendations for school nurses. NASN School Nurse. 2022, 37, 42-7.

"A recent study highlights the importance of building school connectedness in view of COVID-19 and recommends the role of school nurses to help promote school connectedness and ensure students feel safe and supported [52]."